# Not All Splits Are Equal: Rethinking Attribute Generalization Across Dissimilar Categories

Liviu Nicolae Fircă[1,2]    Antonio Bărbălau[1]    Dan Oneata[1,3]    Elena Burceanu[1,3]

[1]Bitdefender, Romania
[2]University of Bucharest, Romania
[3]National University of Science and Technology Politehnica Bucharest, Romania
{ext-abarbalau,doneata,eburceanu}@bitdefender.com

## Abstract

Can models generalize attribute knowledge across semantically and perceptually dissimilar categories? While prior work has addressed attribute prediction within narrow taxonomic or visually similar domains, it remains unclear whether current models can abstract attributes and apply them to conceptually distant categories. This work presents the first explicit evaluation for the robustness of the attribute prediction task under such conditions, testing whether models can correctly infer shared attributes between unrelated object types: *e.g.* identifying that the attribute `has four legs` is common to both DOGS and CHAIRS. To enable this evaluation, we introduce train-test split strategies that progressively reduce correlation between training and test sets, based on: LLM-driven semantic grouping, embedding similarity thresholding, embedding-based clustering, and supercategory-based partitioning using ground-truth labels. Results show a sharp drop in performance as the correlation between training and test categories decreases, indicating strong sensitivity to split design. Among the evaluated methods, clustering yields the most effective trade-off, reducing hidden correlations while preserving learnability of the task (measured through $F_1$ *selectivity* scores). These findings offer new insights into the limitations of current latent representations and inform future benchmark construction for attribute reasoning. Our splits are publicly available on GitHub.

## 1 Introduction

Attributes provide a powerful way for humans to describe objects through shape, color, texture, and taxonomic properties [15], with the compelling ability to transcend class boundaries; for example, the `striped` attribute can be learned from ZEBRAS, BEES, and TIGERS alike. Leveraging this transcendence, Lampert et al. [11] showed that it is possible to classify objects from unseen classes (*e.g.* zero-shot learning) provided that one has their list of attributes at hand. This led Farhadi et al. [6] to propose that image recognition should focus on rich description rather than mere naming, outputting "spotty dog" instead of "dog" and replacing "unknown" with "has four legs and fur."

This ambitious goal necessitates training powerful classifiers to recognize these attributes in ways that generalize to unseen domains or categories of objects. However, existing datasets inadequately evaluate true attribute generalization. Current benchmarks are either taxonomically narrow [11, 18, 25] or fail to control for train-test dissimilarity [9, 14, 21, 26], enabling "semantic leakage" where models exploit taxonomic shortcuts rather than developing genuine attribute abstraction. To address this gap, we introduce dataset splits of increasing difficulty, designed to rigorously assess models' ability to recognize attributes in novel categorical contexts. Our work is related to Attribute Prediction

39th Conference on Neural Information Processing Systems (NeurIPS 2025) Workshop: Reliable ML from Unreliable Data.

and Zero-Shot Learning, Compositional Generalization and Attribute Reasoning Across Dissimilar Categories, which we discuss in Appx. A. Below, we summarize our main contributions:

1. **Evaluating the attribute generalization task**: We are the first to propose an explicit leakage controlled split study for the attribute generalization task. Unlike existing datasets, which evaluate attribute prediction within taxonomically narrow or visually similar domains, we test whether models can abstract attribute knowledge and apply it to unrelated categories that share no superficial similarity with the training set.

2. **Challenging train-test splits to probe generalization**: We introduce a set of novel train-test splits of varying difficulty, based on semantic, perceptual, or taxonomic separation, via LLM-based grouping, embedding similarity, clustering, and supercategory labels. As the correlation between training and test concepts decreases, attribute prediction performance drops significantly, underscoring the impact of split design on generalization.

3. **Clustering achieves minimal leakage without GT labels**: We cluster concept embeddings into groups and assign entire clusters to either the training or test split. Despite being fully unsupervised, this method achieves leakage levels comparable to the ground-truth supercategory-based split, while enabling better generalization performance.

## 2 Split Design for Evaluating Attribute Generalization

We assume a set of **concepts** (*e.g.* CAT, STRAWBERRY, CHAIR), each annotated with binary labels indicating the presence or absence of specific **attributes** (*e.g.* has four legs, tastes good). Our goal is to assess whether these attributes are encoded in distributional representations of the concepts. Examples of such representations include pre-trained embeddings of images depicting the concepts. To quantify the attribute information in the embeddings, we use linear probing [1, 3]: for each attribute we **train a linear classifier** on a subset of concepts and evaluate it on the remaining ones. The standard experimental setup typically involves splitting the concepts randomly in train and test [4, 5, 17]. We introduce **additional splitting strategies**, that explicitly control over the semantic and taxonomic overlap between training and test concepts.

**Our approach.** We group similar concepts using various similarity criteria (detailed below as Grouping methods). These partitioning range from fine-grained ones, with very small groups of similar concepts (*e.g.* LLM-based) to coarse ones, containing very large groups of concepts (*e.g.* Supercategory Labels). Based on these groupings, we assign concepts to either the training or test split, adhering to the following objectives: (a) ensuring similar concepts are placed within the same split, (b) maintaining a comparable positive attribute label rate across splits, and (c) preserving an approximate 80%–20% train-test ratio, with some attributes allowing variation up to a 50%–50%. We describe below the **Grouping methods** we explored:

**Random (RND).** In this split we randomly assign concepts to training and test sets without considering semantic similarity. This is the common approach [4, 5, 17] and it serves as a baseline to assess the degree of leakage tolerated in common evaluation protocols.

**A. LLM-based.** We prompt a LLM (ChatGPT-4o) with the set of concept names and ask it to identify pairs of semantically similar concepts (*e.g.* CUP and MUG). These highly similar pairs are co-assigned to the training set to avoid direct semantic overlap between train and test. The goal is to heuristically reduce leakage through human-like semantic grouping.

**B. Embeddings Similarity.** Given a concept embedding (*e.g.* obtained from pretrained models), we compute the similarity between two concepts as the cosine similarity of their corresponding embeddings. For each concept, we compute the maximum similarity to other concepts and assign the top concepts (with the highest maximum similarity) to the training set. This approach aims to concentrate semantically dense regions in the training set while minimizing high-similarity pairs across the train-test boundary.

**C. Embeddings Clustering.** Given the concepts' embeddings, we also apply K-Means clustering [13] on top. To reduce correlation between splits, entire clusters are assigned to either the training or test set. This approach ensures full coverage of the concept set, as each concept belongs to a cluster.

**GT: Supercategory Labels.** We group the concepts in high-level object categories (superordinate categories or supercategories, in short). For example, BIN and CUP both belong to the "container"

Table 1: **Effect of train-test split strategy on attribute generalization**. Columns represent different split strategies, and rows correspond to various embeddings used as input to the linear probe. Higher correlation with supercategories indicates greater conceptual leakage. Our proposed splits show consistent declines in both metrics across all tested embeddings, offering practical trade-offs between generalization performance and leakage, and providing useful setups for further research on the attribute generalization task.

| LP Features | **SPLITS** ($F_1$ selectivity ↑) | | | | |
| --- | --- | --- | --- | --- | --- |
| | **RND**: Original [17] | **A.** LLM-based | **B.** Similarity | **C.** Clustering | **GT**: Supercategory |
| SigLIP | **45** | 43.7 | 42.8 | 39.9 | 32.1 |
| CLIP | **43.6** | 42.0 | 40.9 | 38.6 | 33.2 |
| Swin-V2 | **43.2** | 42.0 | 39.2 | 34.3 | 25.1 |
| DINOv3 | **40.0** | 38.2 | 36.9 | 34.3 | 27.1 |
| *Correlation with the Supercategory ↓ (mean ± std, detailed in Appx. B)* | | | | | |
| | $0.37 \pm 0.01$ | $0.36 \pm 0.03$ | $0.36 \pm 0.04$ | $0.12 \pm 0.07$ | $\mathbf{0.06} \pm 0.08$ |

supercategory. Each supercategory group is entirely assigned to either the training or testing set, ensuring that no supercategory is shared across splits. This method serves as a strict control to test generalization outside of known taxonomic boundaries.

## 3 Experiments

**Dataset.** We use the McRae×THINGS dataset [17], which contains $1,854$ object concepts (represented as images from THINGS [7]), each annotated with 277 binary attributes (derived from the McRae norms [15]). The dataset presents two main challenges: (1) Label imbalance, as many attributes are rare and require careful train-test splitting to ensure similar positives rates in both sets, and (2) Attribute-supercategory entanglement, where some attributes (*e.g.* has_4_legs) are concentrated within specific supercategories (*e.g.* "mammal"), making it difficult to split without leaking information. So we filtered out attributes that could not be split according to the requirements of **Our approach** (Sec. 2), leaving 211 attributes.

**Experimental Setup.** To represent the concepts, we extract embeddings from vision models, either trained on image-only data (Swin-V2 [12] and DINOv3 [23]) or on image-and-language data (CLIP [20] and SigLIP [27]). For both embedding-based grouping methods, we use Swin-V2 to generate concept embeddings from the THINGS images. As supercategory labels, we use the manual annotations from THINGSplus [24], with 53 supercategories. For linear probing, we used LogisticRegression from scikit-learn [19], with balanced class weights, no regularization, and a maximum of $1,000$ iterations. We have 211 binary classification tasks, one per attribute.

**Evaluation Metrics.** We measure the attribute performance using $F_1$ *selectivity* [8]: the difference between the $F_1$ score and the expected random baseline. We also monitor the *Correlation with the Supercategory (CS)* [17], which measures the extent to which attribute prediction is influenced by supercategory dominance. CS is computed as the Pearson correlation between the per-attribute $F_1$ *selectivity* and the corresponding supercategory dominance (the proportion of positive concepts shared with the best matching supercategory). A high CS score implies reliance on supercategory-specific features, while near-zero indicates minimal dependence. We provide a visualization in Appx. C.

### 3.1 Train-Test Split Design on Attribute Generalization

We investigate how and to what extent conceptual leakage between training and testing affects attribute prediction performance. We evaluate the attribute prediction performance under the five train-test splitting strategies described in Sec. 2. Each method is assessed using $F_1$ *selectivity* and *Correlation with the Supercategory*. The latter serves as a proxy for conceptual leakage: higher correlation indicates stronger reliance on taxonomic shortcuts.

**Results.** We show in Tab. 1 that the random split yields high $F_1$ *selectivity*, but also a high *CS*, suggesting reliance on taxonomic cues rather than true attribute abstraction. The *A. LLM-based* and *B. Embeddings Similarity* splits offer marginal leakage reduction, therefore they still show high correlations. The *GT: Supercategory Labels* split, which is based on GT labels, achieves near-zero

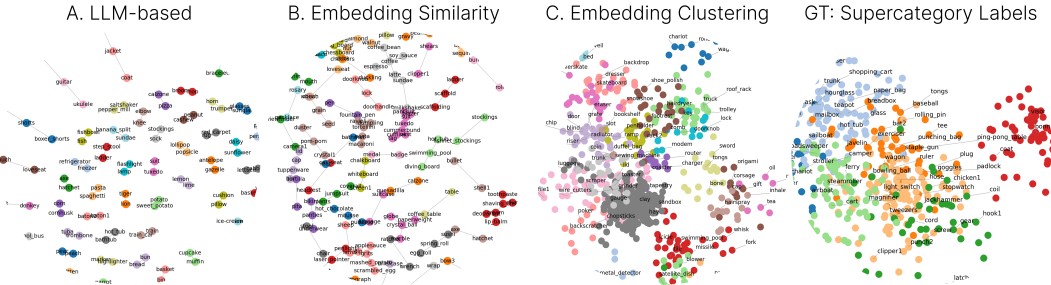

Figure 1: **Granularity and coverage of concepts in the grouping methods**. *A. LLM-based* and *B. Embedding Similarity* offer high precision but leave many concepts ungrouped, risking semantic leakage. While *GT: Supercategory Labels* and *C. Embedding Clustering* both ensure full coverage, the former produces overly broad groups, whereas the latter offers finer granularity, enabling more reliable and controlled train-test splits.

correlation, but at a substantial performance cost, indicating that models struggle to generalize when deprived of taxonomic structure. The *C. Embeddings Clustering* split reduces correlation while retaining significantly better predictive performance. These findings show that current train-test data splits can leak information, leading to biased results in attribute prediction. By introducing splits with different trade-offs between predicting performance and leakages, we provide fairer alternatives.

**Embedding Clustering Ablation.** We observe that the correlation metric remains low across all tested values of $k$ in K-Means ($k \in \overline{10, 400}$), with a maximum correlation of approximately 0.3 across them. Selecting $k = 100$ achieves the most favorable $F_1$ *selectivity* score, while maintaining a *CS* comparable to the lower bound given by the *GT: Supercategory Labels* baseline.

## 3.2 Visualization of Grouping Methods

Fig. 1 illustrates how each method organizes the concepts, emphasizing their defining characteristics and evaluating their influence on the resulting train-test split:

**LLM-based**: Here we have very precise groupings, usually pairs or triplets of semantically similar concepts, covering only 12% of the dataset. The rest remain ungrouped, causing unintended semantic overlap between train and test, as weaker relationships among unassigned concepts are ignored.

**Embedding Similarity**: In this setup, groups are defined based on the top 600 ranked embedding similarities. Although this produces slightly broader groupings than the LLM-based method, the groups remain small. Additionally, approximately 60% of the samples are not assigned to any group.

**Embedding Clustering**: This method is designed to address the shortcomings of the previous approaches. By clustering embeddings into moderately sized groups, it ensures full concept coverage while maintaining sufficient granularity for controlled train-test splitting. This reduces semantic leakage across splits, as reflected in the low correlation scores reported in Tab. 1.

**Supercategory Labels**: This strategy forms broader, more inclusive groupings based on predefined (ground-truth) supercategories. While it ensures that all concepts are assigned to a group, the large group sizes make it difficult to preserve key split properties, such as balanced positive instance rates. In extreme cases, some attributes appear exclusively within a single supercategory.

## 4 Conclusion

We introduced a new benchmark and evaluation protocol for assessing attribute generalization across semantically and perceptually dissimilar categories, settings underexplored in prior work. Our proposed train-test splits vary in difficulty and reveal that generalization performance degrades as semantic overlap between splits decreases, underscoring the importance of split design. Notably, we show that an unsupervised clustering-based split achieves leakage levels comparable to those based on ground-truth labels, while enabling better generalization. Our findings provide a scalable framework for constructing more challenging and realistic attribute prediction benchmarks.

## Acknowledgements

This project has received funding from the European Union's Horizon Europe research and innovation programme under Grant Agreement No: 101120237 (ELIAS).

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

# Appendix

## A   Related Work

**Attribute Prediction and Zero-Shot Learning.** Early work on attribute prediction focused on transferring semantic knowledge across categories via human-defined attributes (*e.g. has tail*, *four-legged*). Datasets such as **Animals with Attributes (AwA)** [11], **SUN Attributes** [18], and **CUB** [25] enabled zero-shot classification by learning attribute classifiers and applying them to novel categories. However, these datasets are taxonomically narrow (*e.g.* all animals or all birds), and generalization often relies on visual or semantic similarity rather than true attribute abstraction.

**Compositional Generalization.** Recent work has explored generalization to unseen (attribute, object) pairs, such as in **MIT States** [9], **UT-Zappos50K** [26], **C-GQA** [14], and **VAW-CZSL** [21]. These benchmarks focus on compositionality, testing whether models can recognize novel (attribute, object) combinations, but do not explicitly control for dissimilarity between training and test concepts. Synthetic datasets such as **CLEVR-CoGenT** [10] enforce disjoint (attribute, object) pairings, but operate in an abstract visual domain.

**Attribute Reasoning Across Dissimilar Categories.** Some works aim to identify shared attributes across semantically distinct objects, such as **CORE** [6] and **Find-the-Common (FTC)** [22]. While aligned in spirit, these datasets are either small-scale or not structured for explicit evaluation of attribute generalization. Methods like **Attributes as Operators** [16] and prompt-based approaches using **CLIP** [2] address compositionality, but do not enforce concept dissimilarity between training and test categories.

## B   Train-Test Split Design on Attribute Generalization (continued)

We present in Tab. 2 the detailed results for the Correlation with the Supercategory metric, across all the embeddings used as input in linear probing.

## C   Visualization of Correlation with the Supercategory

Fig. 2 illustrates, for each attribute, the relationship between its supercategory dominance score (x-axis) and its corresponding $F_1$ *selectivity* (y-axis), as obtained by a linear probe. Each point represents one attribute. The overall trend reflects how much the probe performance correlates with the dominance of a single supercategory in the positive examples.

In the **Random** grouping setting, we observe a clear positive correlation: attributes with concentrated supercategory distributions tend to achieve higher $F_1$ *selectivity*. This suggests that the model may rely on supercategory-specific cues when the split does not explicitly control for semantic leakage.

In contrast, when using groups from **Embedding Clustering**, the points are distributed more uniformly, and the correlation is close to zero. This indicates that the probe performance is less dependent on supercategory dominance, supporting the effectiveness of this split method in reducing unintended information leakage and enforcing better generalization across semantic groups.

Table 2: Effect of train-test split strategy on attribute generalization. Our proposed splits show gradual declines in both metrics, offering practical trade-offs between generalization performance and leakage, and serving as useful setups for further research on the attribute generalization task.

| LP Features | **SPLITS** ($F_1$ *selectivity* ↑) | | | | |
| | **RND**: Original [17] | **A.** LLM-based | **B.** Similarity | **C.** Clustering | **GT**: Supercategory |
|---|---|---|---|---|---|
| SigLIP | **45.0** | 43.7 | 42.8 | 39.9 | 32.1 |
| CLIP | **43.6** | 42.0 | 40.9 | 38.6 | 33.2 |
| Swin-V2 | **43.2** | 42.0 | 39.2 | 34.3 | 25.1 |
| DINOv3 | **40.0** | 38.2 | 36.9 | 34.3 | 27.1 |
| *Correlation with the Supercategory ↓* | | | | | |
| SigLIP | 0.36 | 0.35 | 0.36 | 0.12 | **0.01** |
| CLIP | 0.39 | 0.40 | 0.42 | 0.19 | **0.04** |
| Swin-V2 | 0.36 | 0.35 | 0.32 | 0.02 | **-0.14** |
| DINOv3 | 0.37 | 0.35 | 0.36 | 0.144 | **0.03** |

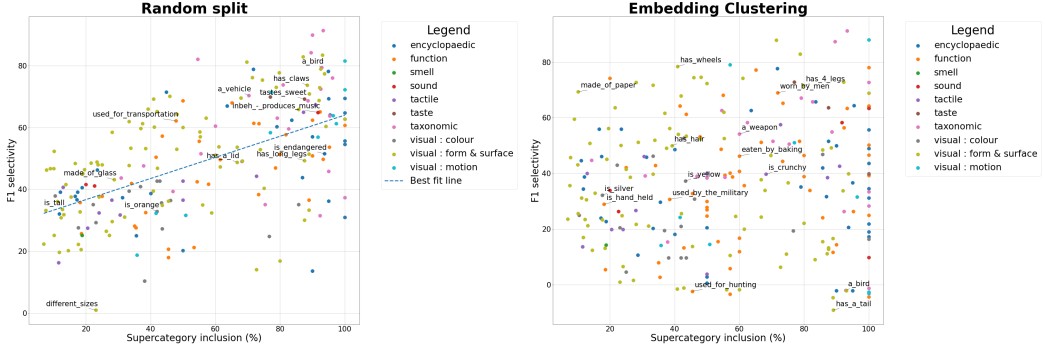

Figure 2: In **Random** grouping (left), a positive correlation emerges, indicating reliance on supercategory-specific features. In contrast, the split based on **Embedding Clustering** yields a near-zero correlation, suggesting improved generalization and reduced semantic leakage.

