# OpenReview forum: "Not All Splits Are Equal: Rethinking Attribute Generalization Across Unrelated Categories"
_NeurIPS.cc/2025/Workshop/Reliable_ML — NeurIPS 2025 - Reliable ML Workshop_

### Official Review · Reviewer_HJf6 · 2025-09-14
**Review of NeurIPS 2025 Workshop Reliable ML Submission80 - Not All Splits Are Equal: Rethinking Attribute Generalization Across Unrelated Categories**

**Rating:** 6
**Confidence:** 2

**Review:**

# Summary
Attribute prediction is the task of learning abstract properties of images beyond object classification, e.g., "striped" is an attribute that can be learned from object classes "ZEBRAS, BEES, and TIGERS".
This paper challenges the standard methods for evaluating attribute generalization in machine learning models, arguing that current benchmarks are flawed by "semantic leakage".
This leakage allows models to exploit taxonomic shortcuts rather than learning to apply abstract attributes (e.g., "has four legs") across conceptually distant categories (e.g., DOGS and CHAIRS). To address this gap, the authors introduce a more rigorous evaluation framework by designing several novel train-test split strategies that reduce the semantic and perceptual correlation between training and test sets. These methods include
1. random (baseline)
1. LLM-based grouping,
1. embedding similarity,
1. embedding clustering, and
1. a strict ground-truth supercategory partition.

Experimentally, the authors demonstrate that as the correlation between training and test sets increases, attribute prediction performance drops sharply, confirming that standard evaluations overestimate true generalization. Among the proposed methods, an unsupervised
embedding clustering approach emerged as the most effective trade-off.

# Strengths
- The paper's primary strength is its novel contribution to evaluating attribute generalization. It is the first work to explicitly design and test train-test splits that control for semantic and perceptual similarity between categories. The proposed splitting strategies, particularly the unsupervised clustering method, offer a new and scalable framework for creating more challenging benchmarks.
- The proposed empirical evaluation is methodologically rigorous. The authors systematically compare five distinct splitting strategies, from a standard random baseline to a strict ground-truth partition. This comparison is conducted across four different pre-trained embedding types (SigLIP, CLIP, Swin-V2, DINOV3), demonstrating that the observed performance drop is a consistent phenomenon and not an artifact of a single model.

# Weaknesses
- The LLM-based and embedding similarity-based methods are somewhat problematic since they fail to assign all points to a group. It's not clear to me if this is an inherent problem or if the correlation/performance tradeoffs can be improved by fixing this issue.
- The conclusions are based on a single dataset.
- The goal is to reduce correlation between train/test sets, since this might overinflate the generalization results. However, it seems like embedding similarity achieves worse generalization even with higher correlation compared to embedding clustering. This makes me curious if the correlation between train/test sets is really the underlying cause of overinflated test results.

# Suggestions
1. Could the authors explain why the LLM-based and embedding similarity-based methods fail to assign all points to a group? Could a variation of their algorithm fix this issue?
1. The findings would be more convincing if the experiments were completed on multiple datasets
1. Could the authors explain why embedding similarity achieves worst generalization despite higher correlation compared to embedding clustering?